

# Are mountain habitats becoming more suitable for generalist than cold-adapted lizards thermoregulation?

Zaida Ortega,  Abraham Mencía and  Valentín Pérez-Mellado

Department of Animal Biology, University of Salamanca, Salamanca, Spain

## ABSTRACT

Mountain lizards are highly vulnerable to climate change, and the continuous warming of their habitats could be seriously threatening their survival. We aim to compare the thermal ecology and microhabitat selection of a mountain lizard, *Iberolacerta galani*, and a widely distributed lizard, *Podarcis bocagei*, in a montane area. Both species are currently in close syntopy in the study area, at 1,400 m above the sea level. We determined the precision, accuracy and effectiveness of thermoregulation, and the thermal quality of habitat for both species. We also compared the selection of thermal microhabitats between both species. Results show that *I. galani* is a cold-adapted thermal specialist with a preferred temperature range of 27.9–29.7 °C, while *P. bocagei* would be a thermal generalist, with a broader and higher preferred temperature range (30.1–34.5 °C). In addition, *I. galani* selects rocky substrates while *P. bocagei* selects warmer soil and leaf litter substrates. The thermal quality of the habitat is higher for *P. bocagei* than for *I. galani*. Finally, *P. bocagei* achieves a significantly higher effectiveness of thermoregulation (0.87) than *I. galani* (0.80). Therefore, these mountain habitat conditions seem currently more suitable for performance of thermophilic generalist lizards than for cold-specialist lizards.

Corresponding author
Zaida Ortega, zaidaortega@usal.es

## INTRODUCTION

Climate change has already produced several impacts in the biology and distribution of many animal species worldwide (*Parmesan*, *2006*; *McCain*, *2010*). However, some species are more vulnerable than others to the impact of global warming (*Araújo, Thuiller & Pearson*, *2006*; *Sinervo et al.*, *2010*; *Huey et al.*, *2012*). Ectotherms are particularly sensitive to climate warming since they depend on external heat sources for body temperature upkeep (e.g., *Hertz, Huey & Stevenson*, *1993*; *Huey et al.*, *2012*). Knowledge on the thermal biology of ectotherms is necessary to assess their vulnerability to climate change, to predict the future impacts of global warming, and to adopt conservation measures to prevent their extinction (*Carvalho et al.*, *2010*; *Crossman, Bryan & Summers*, *2012*; *Groves et al.*, *2012*).

High elevation ectotherms would be particularly threatened by the fast increase of environmental temperatures, mainly because two reasons: (1) the plasticity and evolution of thermal physiology seem limited to keep pace with the fast environmental warming

(e.g., *Muñoz et al.*, *2014*; *Gunderson & Stillman*, *2015*), and (2) living in mountaintops, they lack colder areas to migrate (*Araújo, Thuiller & Pearson*, *2006*; *Berg et al.*, *2010*; *McCain*, *2010*). In addition, mountain species tend to be cold-specialists (e.g., *Aguado & Braña*, *2014*), which makes them more vulnerable, because the decline of fitness when body temperatures exceed the optimum temperature is greater in narrower thermal reaction norms (*Martin & Huey*, *2008*; *Huey et al.*, *2012*; *Gunderson & Stillman*, *2015*). Mountain lizards could also be threatened by potential displacement by thermal generalist species with a broader distribution at the surrounding lowlands that may ascend in altitude as warming increases (*Araújo, Thuiller & Pearson*, *2006*; *Huey et al.*, *2012*; *Comas, Escoriza & Moreno-Rueda*, *2014*). An expansion, both in altitude and latitude, due to climate change has already been documented for several species (*Parmesan*, *2006*; *Sinervo et al.*, *2010*; *Chen et al.*, *2011*; *Moreno-Rueda et al.*, *2012*; *Bestion, Clobert & Cote* , *2015*). These factors, altogether, would place high-mountain lizards among the most vulnerable animals worldwide, especially mountain lizards of the Iberian Peninsula, due to the higher warming and drought predicted for these areas (*Nogués-Bravo et al.*, *2008*; *Araújo et al.*, *2011*; *Maiorano et al.*, *2013*). The study of the thermal ecology of these mountain lizards, as well as the comparison with their potential competitors, would be useful to design the conservation measures required to preserve the species (*Urban, Tewksbury & Sheldon*, *2012*; *Lord & Whitlatch*, *2015*).

Some studies had assessed thermal ecology of other *Iberolacerta* lizards (*Monasterio et al.*, *2009*; *Aguado & Braña*, *2014*; *Ortega, Mencía & Pérez-Mellado*, *2016*). We studied the thermal ecology of the León rock lizard, *I. galani*, a mountain lizard living in its historical range, and the Bocage's wall lizard, *P. bocagei*, that has expanded its altitudinal range to the study area in recent years. Both are medium-size, insectivorous and heliothermic lacertid lizards, endemic from the northwestern of Spain. Their distribution ranges are considerably different (Fig. 1): *I. galani* is restricted to high-mountain climate isolated areas from 1,300 to 2,500 m asl (meters above the sea level; *Arribas, Carranza & Odierna*, *2006*; *Mencía, Ortega & Pérez-Mellado*, *2016*), whereas *P. bocagei* inhabits a variety of habitats from the sea level to 1,900 m asl habitats (*Galán*, *1994*; *Pérez-Mellado*, *1998*; *Galán*, *2004*). However, both live in close syntopy in the study area nowadays, fully mixed in the same habitat. We compared the thermal requirements of the two species in the laboratory and the thermal traits of their habitat in order to study if mountain habitats are increasingly unsuitable for mountain lizards, and may be favoring the expansion of thermal generalists instead. We first aim to assess and compare the thermal preferences and behavioral thermoregulation of both species (*Hertz, Huey & Stevenson*, *1993*; *Angilletta*, *2009*). Then, we studied their selection of microhabitats and we compared the thermal suitability of the habitat for both species under the current climatic conditions.

## MATERIALS & METHODS

### Study area

The study area was in the Natural Monument ''Lago de La Baña'' (León province, Spain; 42°15′N, 6°29′W). It was an area surrounding a glacial lake, located at 1,400 m asl, formed by
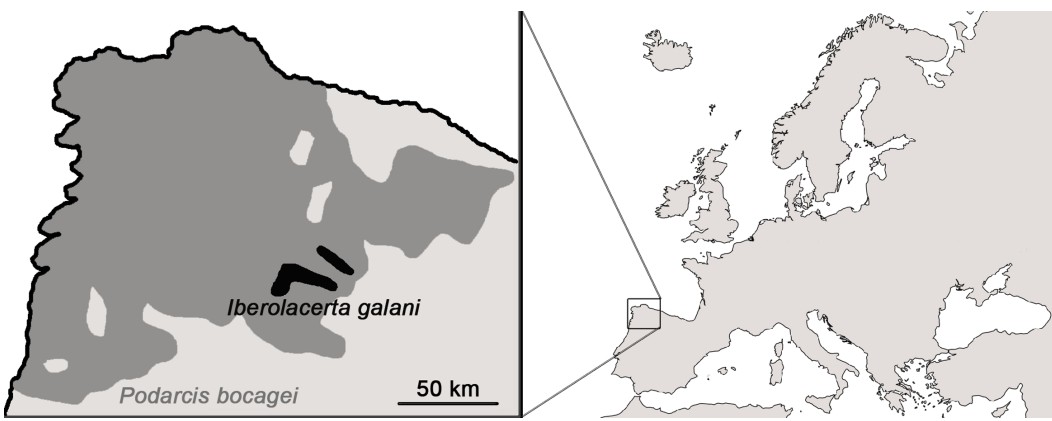

**Figure 1 Distributional ranges of *Iberolacerta galani* and *Podarcis bocagei*.** *I. galani* is restricted to the mountain range of Montes de León (Spain) while *P. bocagei* is widely distributed in the northwest of the Iberian Peninsula.

**Table 1 Size and mass of both species.** Snout-vent length (SLV, in mm) and weight (in g) of adult males and females of *Iberolacerta galani* and *Podarcis bocagei* of La Baña (León, Spain) included in the study.

|  |  | SVL | Weight |
|---|---|---|---|
| *I. galani* | Males ($n = 44$) | 65.95 ± 5.83 | 6.94 ± 1.87 |
|  | Females ($n = 53$) | 67.96 ± 6.07 | 7.02 ± 1.83 |
| *P. bocagei* | Males ($n = 48$) | 58.52 ± 4.89 | 4.90 ± 1.27 |
|  | Females ($n = 25$) | 55.38 ± 4.66 | 3.86 ± 1.08 |

slate rocks, meadows and shrubs. The area is also circled by mountains peaks of more than 2,000 m asl on one side and deteriorated slate quarries in the other side (*Fuertes-Gutiérrez & Fernández-Martínez*, *2010*).

## Field sampling

Body temperatures ($T_b$) and operative temperatures ($T_e$) were recorded simultaneously in the field during August 2011, 2012 and 2013, in order to avoid the effect of seasonal variations. Length and weight of the studied animals are provided in Table 1. Operative temperatures ($T_e$) estimate the temperatures that non-thermoregulating lizards would reach, if they were distributed randomly in their habitat (*Hertz, Huey & Stevenson*, *1993*). For recording $T_e$, we employed copper models of the same size of lizards (*Bakken & Angilletta*, *2014*). One thermocouple probe was placed into each hollow model and connected to a data logger HOBO H8 (®Onset Computer Corporation) programmed to take a temperature record every five minutes. The data loggers with models were placed in different microhabitats: rock (in different orientations), soil, log, leaf litter, and grass. For measuring $T_b$, we captured lizards by noosing during their daily activity period, from 08.00 to 18.00 h GMT (Greenwich Mean Time). On each capture, we measured cloacal body temperature ($T_b$) immediately after capture (<30 s after capture) with a Testo® 925 digital thermometer (±0.1 °C precision). We also registered the time of the day, the type of substrate, the distance from the nearest potential refuge and the height of the microhabitat from the floor.

For 30 individual lizards (15 *I. galani* and 15 *P. bocagei*), we also recorded these variables ($T_a$, $T_s$, time of the day, type of microhabitat, height of the point from the floor, and distance to the nearest potential refuge) at four points associated to each capture place, in order to get an approach of the habitat structure. Each point was 1 m away from the capture site in the direction of the main four cardinal points (N, S, E, and W). Thus, the measures at random points represent the availability of all variables in the habitat, in order to compare with the values in the microhabitats used by both species. We refer to the values of these random points as 'availability' in the results section.

Lizards were sampled under licences of the Castilla y León Environmental Agency (EP/CYL/320/2012). The study was conducted in compliance with all ethical standards and procedures of the University of Salamanca.

### Preferred temperature range

Thermal preferences of lizards in the laboratory represent the body temperatures that lizards would achieve in their habitats in the absence of other ecological restrictions but temperature (e.g., *Hertz, Huey & Stevenson*, *1993*; *Angilletta*, *2009*). The preferred temperature range of *I. galani* was measured in August 2011 and the preferred temperature range of *P. bocagei* was measured in August 2013. All conditions were replicated for both species: field area of capture of lizards, laboratory conditions and materials (terraria, thermometer and lamp), as well as the methodology (thermal gradient and the protocol of measurement). Lizards were housed in individual terraria, fed daily with mealworms (*Tenebrio molitor*) and crickets (*Gryllus assimilis*), and provided with water *ad libitum.* The thermal gradient was built in a glass terrarium ($100 \times 60 \times 60$ cm) with a 150 W infrared lamp over one of the sides, obtaining a gradient between 20 and 60 °C. A data of a preferred body temperature ($T_{\text{pref}}$) of a lizard was recorded in the cloaca with a digital thermometer (Testo® 925) each hour in the period from 08.00 to 18.00 h (GMT). We used 24 adult lizards (12 males, 12 females) from each species, with 6 hourly measures of $T_{\text{pref}}$ of each individual lizard. The 50% of central values of preferred body temperatures (that is, the interquartile range) was considered as the preferred temperature range to assess thermoregulation, as this is a common metric used to assess thermoregulation (*Hertz, Huey & Stevenson*, *1993*; *Blouin-Demers & Nadeau*, *2005*). After both experiments, lizards were released completely unharmed at their capture sites.

### Indexes of thermoregulation

To test the null hypothesis of thermoregulation (that is, if lizards use microhabitats randomly regarding temperature) we followed the protocol developed by *Hertz, Huey & Stevenson* (*1993*), and calculated three indexes of thermoregulation. The first is the index of accuracy of thermoregulation ($\bar{d}_b$), that is the mean of absolute values of the deviations between each $T_b$ from the preferred temperature range. Thus, the values of the index of accuracy of thermoregulation are counterintuitive: higher values of $\bar{d}_b$ indicate lower accuracy of thermoregulation, and vice-versa. The second is the index of thermal quality of habitat ($\bar{d}_e$), calculated as the mean of absolute values mean of the deviations of each $T_e$ from the preferred temperature range. Accordingly, the values of the index
of thermal quality of the habitat are also counterintuitive: higher values of $\bar{d}_e$ indicate a lower thermal quality of the habitat, and vice-versa. The third is the index of effectiveness of thermoregulation ($E$), that is calculated as $E = 1 - \bar{d}_b/\bar{d}_e$. Hence, values of $E$ range from 0 to 1, meaning a higher effectiveness of thermoregulation as higher is the value of $E$ (see *Hertz, Huey & Stevenson*, 1993). The effectiveness of thermoregulation was calculated with THERMO, a Minitab module written by Richard Brown. THERMO has been used in previous studies of thermal ecology (e.g., *Ortega et al.*, 2014) and uses three kinds of input data: $T_b$, $T_e$ and $T_{\text{pref}}$ of the preferred temperature range, and was programed to perform bootstraps of 100 iterations, building pseudo-distributions of three kinds of output values: the arithmetic mean of the index of accuracy of thermoregulation ($\bar{d}_b$), the arithmetic mean of the index of thermal quality of the habitat ($\bar{d}_e$), and the arithmetic mean of the index of effectiveness of thermoregulation ($\bar{E}$).

## Data analysis

All means were reported with standard deviations (sd). Parametric statistics were performed when data followed the assumptions of normality and variance homogeneity. If these assumptions were not fulfilled, even after log-transformation, non-parametric equivalents were carried out (*Crawley*, 2012; *Sokal & Rohlf*, 1995). Analyses were conducted using R, version 3.1.3 (*R Core Team*, 2015). Post-hoc comparisons of Kruskal–Wallis tests were computed with Nemenyi test with the package PMCMR (*Pohlert*, 2014).

## RESULTS

### Temperatures of the habitat

There were significant differences between the $T_e$ offered by the different microhabitats (Kruskal–Wallis test, $H = 2669.642, df = 15, P < 0.0001, n = 6082$). According to the results of the post-hoc comparisons of their $T_e$, the microhabitats could be classified into four groups: (1) cold microhabitats that were below the preferred temperature range of both species, as would be south-facing rock in full sun, and flat rock, soil and the leaf litter in shade, (2) mild microhabitats that provided $T_e$ within the preferred temperature range of both species, as would be under rock microhabitats, north-facing and the east-facing rock in full sun, and flat rock, soil and the leaf litter in filtered sun, at some hours of the day, (3) warm microhabitats that provided $T_e$ that exceeded the preferred temperature range of *I. galani* but felt within the preferred temperature range of *P. bocagei* during some hours of the day, as would be the microhabitats of flat, east-facing and west-facing rock, and log in full sun, and (4) very warm microhabitats that exceeded the preferred temperature range of both species during all day, as is the case of grass, log, leaf litter, and soil in full sun (see Fig. 2).

### Microhabitat selection

Both species selected microhabitats not-randomly regarding $T_a$, being the mean $T_a$ of capture places of lizards higher than the mean $T_a$ available in the habitat (Fig. 3). However, $T_a$ selected by *P. bocagei* were significantly higher than those selected by *I. galani* (Fig. 3). In addition, both species selected microhabitats with similar $T_s$, higher than that available

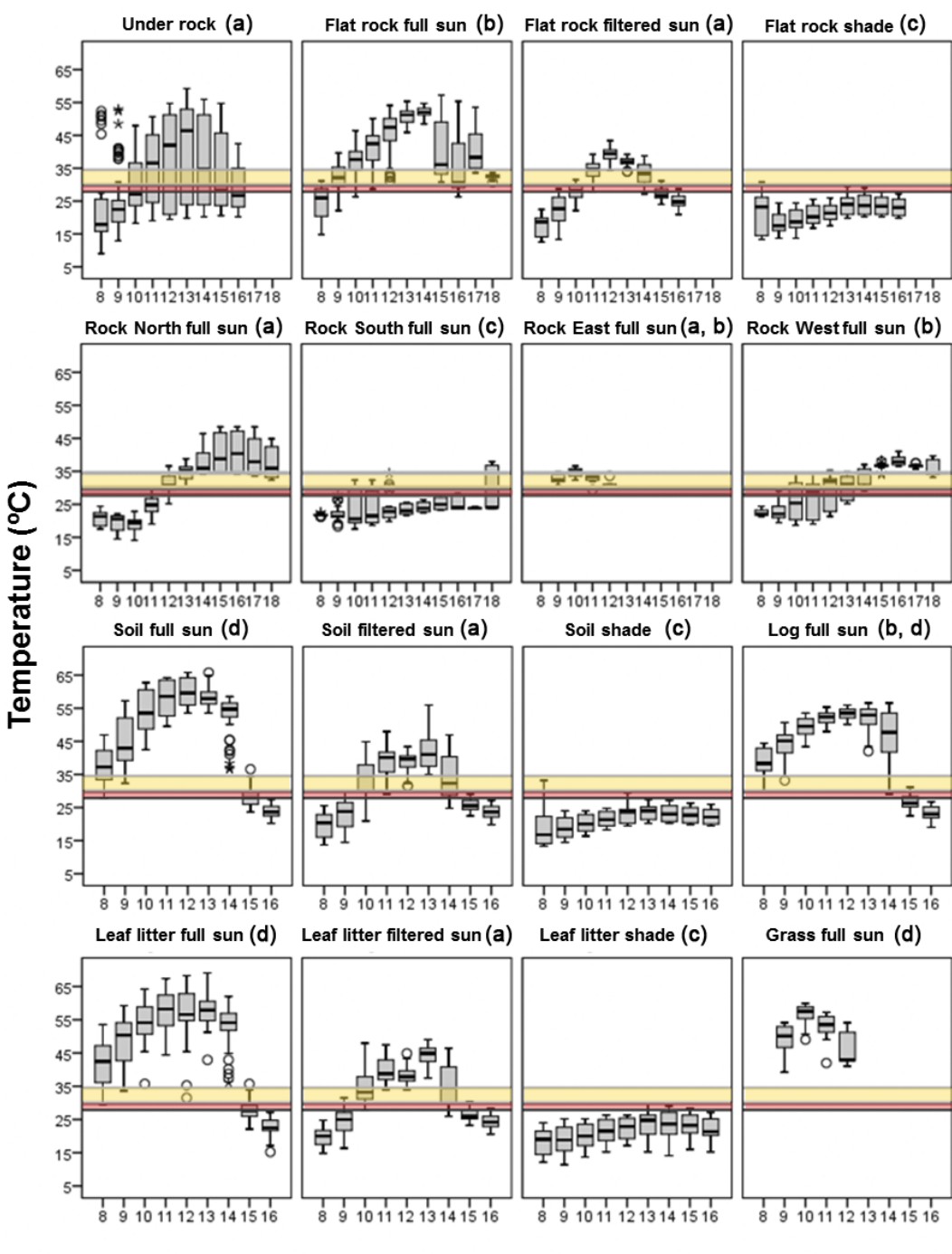

**GMT Hour**

**Figure 2  Operative temperatures.** Boxplots of the operative temperatures of the different microhabitats of the study area during the daily activity period of lizards. Horizontal bands represent the 50% preferred temperature range of *Iberolacerta galani* (red band) and *Podarcis bocagei* (yellow band). The different letters indicate significant different operative temperatures among microhabitats in the post-hoc paired comparisons of Kruskal Wallis.

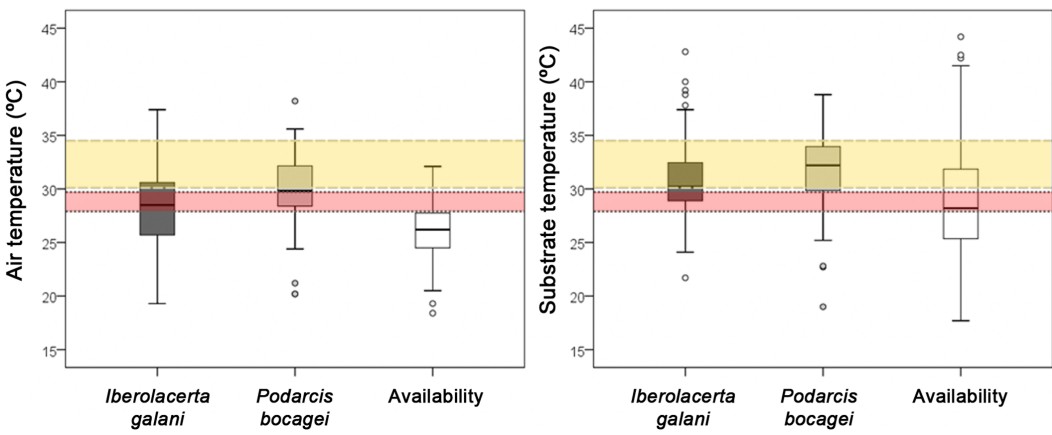

**Figure 3** **Comparison of the environmental temperatures of the selected microhabitats of both species and the mean availability of the habitat.** Boxplots of air and substrate temperature of the capture locations of *Iberolacerta galani* and *Podarcis bocagei*, and the general availability of the habitat. Both species selected microhabitats with greater mean air temperature than the mean air temperature randomly available in the habitat (all post-hoc paired comparisons of Kruskal–Wallis are significant, for the paired comparison of *I. galani-P. bocagei* regarding substrate temperature). Horizontal bands represent the 50% preferred temperature range of *Iberolacerta galani* (red band) and *Podarcis bocagei* (yellow band).

(Fig. 3). Regarding the distance to the nearest potential refuge, both species selected microhabitats that were closer to potential refuges than the mean habitat availability (*I. galani*: mean distance $= 13.56 \pm 11.52$ cm, $n = 66$; *P. bocagei*: mean distance $= 16.04 \pm 17.24$cm, $n = 72$; availability: mean $= 39.76 \pm 31.30$ cm, $n = 123$; Kruskal–Wallis test, $H = 62.198$, $df = 2$, $P < 0.0001$, post-hoc comparisons were significant only for: *I. galani*-availability, $P < 0.0001$, and *P. bocagei*- availability, $P < 0.0001$). *I. galani* selected microhabitats higher than the mean availability, while *P. bocagei* selected microhabitats of similar height of the mean available height (*I. galani*: height $= 11.17 \pm 18.24$ cm, $n = 78$; *P. bocagei*: height $= 7.57 \pm 16.70$ cm, $n = 72$; availability $= 10.30 \pm 22.17$ cm, n $= 123$; Kruskal–Wallis test, $H = 15.824$, $df = 2$, $P < 0.0001$, *post-hoc* comparisons were significant only for: *I. galani*- availability, $P = 0.009$, and *I. galani-P.bocagei*, $P < 0.0001$).

Both species clearly selected microhabitats with a significantly different frequency than the abundance of each type of microhabitat (Fig. 4). *I. galani* selected microhabitats with a smaller presence of grass than the randomly available in the habitat (Fisher exact test, $P < 0.0001$). Meanwhile, *P. bocagei* also selected soil microhabitats with a greater proportion than the randomly available in the habitat (Fisher exact test, $P < 0.0001$; Fig. 4). Finally, there were statistically significant differences between the two species in the selection of microhabitats (Fisher exact test, $P < 0.0001$), especially regarding the higher selection of rocky areas by *I. galani* (Fig. 4). Table 2 shows the proportion of $T_e$ of the different types of microhabitat that felt below, within and above the preferred temperature range of each species.

## Lizard thermoregulation

There were no differences between males and females in the average preferred temperatures, neither in *P. bocagei* (males: $\bar{T}_{pref} = 32.9 \pm 1.63\,°C$, $n = 12$; females: $\bar{T}_{pref} = 31.8 \pm 2.44\,°C$,
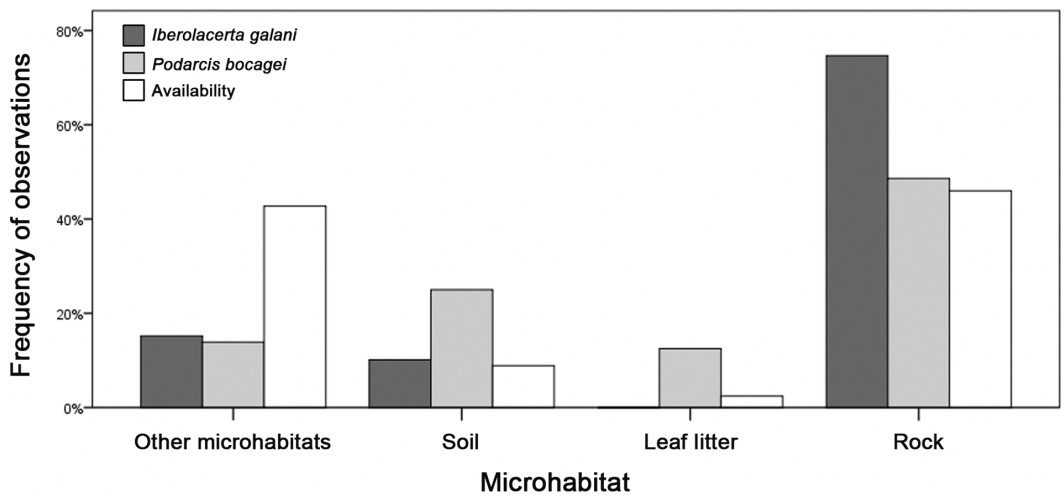

**Figure 4** **Microhabitat preferences of both species.** Microhabitats where *Iberolacerta galani* and *Podarcis bocagei* lizards occurred and availability of the different types of microhabitats measured though random points, indicated as percentage of observations for each category. The category of 'other microhabitats' includes grass and logs.

**Table 2** **Thermal suitability of studied microhabitats for both species.** Proportion (%) of the operative temperatures ($T_e$) of the different studied microhabitats that are lower, within, of higher than the preferred temperature range (PTR) of *Iberolacerta galani* and *Podarcis bocagei* during summer in the study. The microhabitat category of 'other' includes grass and logs. All microhabitats were measured under all sun situations (full sun, filtered sun and shade), so the sun situation is homogeneous among them, making the proportions comparable.

|  |  | Other | Soil | Leaf litter | Rock | Total |
|---|---|---|---|---|---|---|
| *I. galani* | Lower | 9.8% | 52.5% | 48.4% | 46.1% | 46.9% |
|  | Within | 1.7% | 3.5% | 6.2% | 6.4% | 5.1% |
|  | Higher | 88.5% | 44.0% | 45.4% | 47.5% | 48.0% |
| *P. bocagei* | Lower | 11.8% | 56.7% | 55.4% | 53.8% | 52.9% |
|  | Within | 1.6% | 4.1% | 4.3% | 4.0% | 8.5% |
|  | Higher | 86.6% | 39.2% | 40.3% | 38.7% | 38.6% |

$n = 12$; ANOVA, $F_{1,22} = 1.746$, $P = 0.200$), nor in *I. galani* (males: $\bar{T}_{\text{pref}} = 28.9 \pm 0.48$ °C, $n = 12$; females: $\bar{T}_{\text{pref}} = 28.8 \pm 0.49$ °C, $n = 12$; ANOVA, $F_{1,22} = 0.118$, $P = 0.734$). Thus, data of both genders were combined in subsequent analyses. The average preferred temperatures were lower for *I. galani* than for *P. bocagei* (*I. galani*: $\bar{T}_{\text{pref}} = 28.8 \pm 0.47$ °C, $n = 24$; *P. bocagei*: $\bar{T}_{\text{pref}} = 32.3 \pm 2.11$ °C, $n = 24$; Mann–Whitney $U$-test, $U = 42.00$, $P < 0.0001$). Thus, the preferred temperature range of *I. galani* was 27.9–29.7 °C and the preferred temperature range of *P. bocagei* was 30.1–34.5 °C. Furthermore, the preferred temperature range of *I. galani* was significantly narrower than the preferred temperature range of *P. bocagei* (Levene's test, $W = 33.151$, $P < 0.0001$).

In the field, *I. galani* exhibited lower $T_b$ than *P. bocagei* (*I. galani*: $\bar{T}_b = 30.9 \pm 2.39$ °C, $n = 79$; *P. bocagei*: $\bar{T}_b = 33.9 \pm 3.03$ °C, $n = 72$; ANOVA, $F_{1,149} = 45.061$, $P < 0.0001$; Fig. 5). The index of thermal quality of habitat ($\bar{d}_e$) was significantly higher for *I. galani* than for *P. bocagei* (Mann–Whitney $U$-test, $U = 24.0$, $P < 0.0001$; Table 3). In addition,

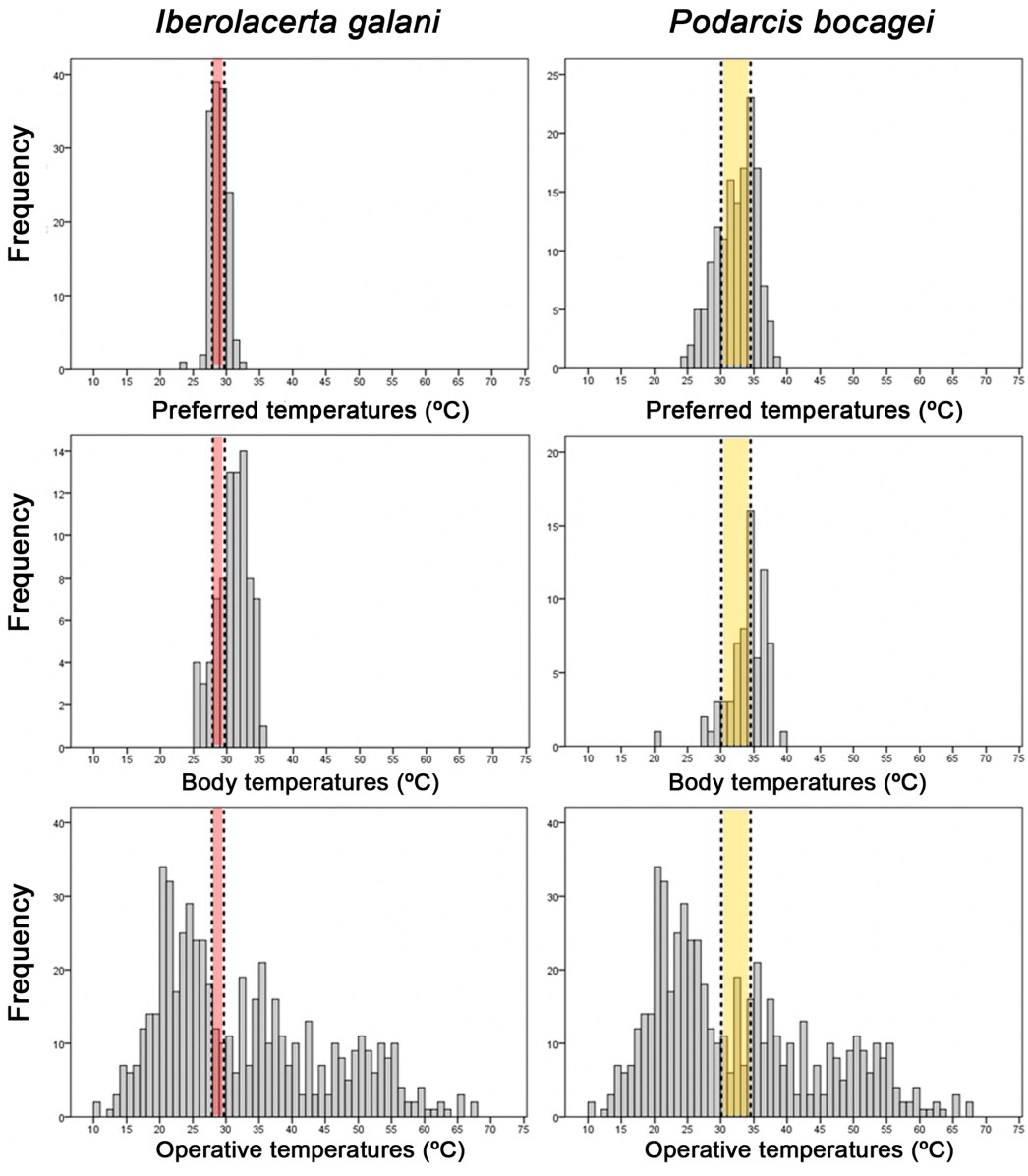

**Figure 5** **Themoregulation of both species.** Histograms of preferred temperatures, body temperatures and operative temperatures of *Iberolacerta galani* and *Podarcis bocagei* lizards. Horizontal bands represent the 50% preferred temperature range of *Iberolacerta galani* (red band) and *Podarcis bocagei* (yellow band).

*I. galani* showed a higher value of the index of thermoregulation accuracy ($\bar{d}_b$; ANOVA, $F_{1,198} = 1086.86$, $P < 0.0001$; Table 3). Finally, *I. galani* achieved a lower effectiveness of thermoregulation (Mann–Whitney $U$-test, $U = 77.0$, $P < 0.0001$; Table 3).

## DISCUSSION

The preferred temperature range of a species is assumed to approximately reflect the optimum range for fitness (*Hertz, Huey & Stevenson*, *1993*; *Martin & Huey*, *2008*). *I. galani* has a low and very narrow preferred temperature range (27.9–29.7 °C), being the lowest found

**Table 3 Indexes of thermoregulation of both syntopic populations.** Mean (±sd) values of the indexes of thermal quality of the habitat ($d_e$), accuracy of thermoregulation ($d_b$) and effectiveness of thermoregulation ($E$) for *Iberolacerta galani* and *Podarcis bocagei* living in close syntopy. Values of the 95% CI of each index are provided in brackets.

| | $\bar{d}_e$ (°C) | $\bar{d}_b$ (°C) | $\bar{E}$ |
|---|---|---|---|
| *I. galani* | 9.36 ± 0.03 (9.294/9.419) | 1.88 ± 0.02 (1.848/1.912) | 0.80 ± 0.002 (0.795/0.802) |
| *P. bocagei* | 8.30 ± 0.02 (8.249/8.347) | 1.05 ± 0.02 (1.012/1.089) | 0.87 ± 0.002 (0.869/0.878) |

to date in Lacertidae (*Bauwens et al.*, 1995; *Aguado & Braña*, 2014; *Ortega, Mencía & Pérez-Mellado*, 2016). By contrast, the preferred temperature range of *P. bocagei* (30.1–34.5 °C) is significantly higher and wider than that of *I. galani*, and both ranges do not even overlap. Thus, our data suggest that *I. galani* would be a cold-adapted thermal specialist species, like other species of the genus *Iberolacerta* (*Martín & Salvador*, 1993; *Aguado & Braña*, 2014; *Žagar et al.*, 2015; *Ortega, Mencía & Pérez-Mellado*, 2016), while *P. bocagei* would be a thermal generalist with preference for warmer temperatures, like other species of the genus *Podarcis* (*Bauwens et al.*, 1995; *Bauwens, Hertz & Castilla*, 1996; *Capula et al.*, 2014; *Ortega et al.*, 2014). Environmental temperatures are predicted to continuously rise in the mountains of the Iberian Peninsula during the coming years (*Araújo, Thuiller & Pearson*, 2006; *Nogués-Bravo et al.*, 2008) and the difference between being a thermal specialist or a generalist will be crucial when determining the vulnerability of a given species to climate change (*Martin & Huey*, 2008; *Huey et al.*, 2012). Thermal reaction norms of fitness are asymmetric: fitness gradually increases from the critical minimum temperature up to the physiological optimum, just to decline sharply when body temperature exceeds the physiological optimal temperature (*Huey & Stevenson*, 1979; *Angilletta, Huey & Frazier*, 2010). Due to the asymmetry of the thermal reaction norm curve, an increase in body temperatures exceeding the optimal temperature leads to a higher reduction of fitness than a similar decrease in body temperatures, as predicted by the Jensen's inequality (*Martin & Huey*, 2008; *Huey et al.*, 2012). Moreover, this negative effect of exceeding the optimal temperature is higher the more specialized the species is (*Huey et al.*, 2012). Consequently, not only *I. galani* preferred temperature range is lower than that of *P. bocagei* and the increase of environmental temperatures will exceed it before, but also their narrower preferred temperature range suggest that the negative effects of exceeding their optimal temperatures would be more detrimental to *I. galani* (*Martin & Huey*, 2008; *Huey et al.*, 2012).

Given a scenario of continued warming, there are three alternatives for mountain lizards: to adapt, to shift their ranges, or to extinguish (*Berg et al.*, 2010; *Gunderson & Stillman*, 2015). The ability of *I. galani* to disperse is limited by the peaks of mountains, so that, sooner or later, lizards migrating upwards to avoid overheating would run out of space to migrate (*Araújo, Thuiller & Pearson*, 2006; *Huey et al.*, 2012). Therefore, mountain lizards would only avoid extinction by adapting to warmer environments. It was recently discovered that the plasticity of the critical temperatures has little capacity for change, even lesser for the critical thermal maximum, so it will probably not be enough to keep pace with the rate of environmental warming (*Gunderson & Stillman*, 2015). However, the ability of ectotherms to behaviorally buffer environmental temperature changes could

probably mitigate the negative impact of global warming in fitness of mountain lizards for a while (*Kearney, Shine & Porter*, *2009*; *Huey et al.*, *2012*). Their high effectiveness of thermoregulation and their ability to select the most suitable microhabitats, suggest that *I. galani* may have the capacity for behavioral buffering of the impact of climate warming. Furthermore, the habitat under study is heterogeneous, with a mosaic of microhabitats offering different operative temperatures, some colder, some warmer, and some equal to the preferred temperature range of this species. Hence, the habitat provides *I. galani*, at least for some time, the possibility to use its behavioral adjustments to buffer the impact of global warming (*Huey et al.*, *2012*; *Sears & Angilletta*, *2015*). It would be also possible that the adaptation of the thermal preferences would contribute to the buffering of the temperature rising, at least for a while (e.g., *Gvoždík*, *2011*).

*P. bocagei* select microhabitats of soil and leaf litter with higher frequency than the randomly available in its habitat, which are significantly warmer than the rocky substrates preferred by *I. galani*. Furthermore, the proportion of operative temperatures fitting the preferred temperature range of *P. bocagei* exceeds nowadays that fitting the preferred temperature range of *I. galani*. In general, air temperature diminishes as elevation rises (e.g., *Körner*, *2007*). Nonetheless, this montane habitat is, at present, more suitable for *P. bocagei* than for *I. galani*. In addition, the warm habitat condition leads *I. galani* lizards to achieve a lower effectiveness of thermoregulation than *P. bocagei*, which may entail less physiological performance and fitness for *I. galani*. Hence, our results indicate that the thermophilic species has taken a better advantage of the current thermal environment than the cold-adapted species, maybe favoured by climate warming. Moreover, generalist lizard species can efficiently exploit high-elevation cold habitats, by means of different adaptations (*Zamora-Camacho, Reguera & Moreno-Rueda*, *2015*), which in this situation could increase *I. galani* competitive exclusion risk. *Monasterio et al.* (*2009*) reported a higher effectiveness of thermoregulation of *Iberolacerta cyreni* than *Podarcis muralis* in high mountain areas, a situation that should be the usual but maybe reverting, as we report here, given that the mountain habitats are becoming warmer. However, we do not know if *I. galani* and *P. bocagei* compete for food, refuges or any other resources. Thus, the generalist could expand without the specialist being affected, unless the specialist would be compromised by the new conditions, and that would not be qualified as displacement, instead, the two species would be reacting independently to climate change.

*Iberolacerta* lizards appear to have not adapted to warm conditions in the past, after the last glaciation and, consequently, they were relegated to areas of higher altitude (*Carranza, Arnold & Amat*, *2004*; *Crochet et al.*, *2004*; *Mouret et al.*, *2011*). Hence, it is unlikely that they will be able to cope with the current climatic change, which entails faster warming than the species have ever met before (*Diffenbaugh & Field*, *2013*). On the contrary, *P. bocagei* lizards remained in warm refugia during past cold periods, and probably colonized their current distribution within last 10000 years (*Pinho et al.*, *2011*). In short, we are probably documenting, with human induced climate change, a remake of past biogeographic spread of generalist lizard species and the concomitant restriction of cold-adapted species to colder areas.

## ACKNOWLEDGEMENTS

We thank the people of La Cabrera Baja for their hospitality during the fieldwork periods. We also thank Mary Trini Mencía and Joe McIntyre for linguistic revision, and Mario Garrido, Ana Pérez-Cembranos, Gonzalo Rodríguez and Alicia León for support during the writing process.

### Funding

Financial support was provided to ZO and AM by predoctoral grants of the University of Salamanca (FPI program). This work was also supported by the research projects CGL2009-12926-C02-02 and CGL2012-39850-CO2-02 from the Spanish Ministry of Science and Innovation. The funders had no role in study design, data collection and analysis, decision to publish, or preparation of the manuscript.

### Grant Disclosures

The following grant information was disclosed by the authors:
University of Salamanca.
Spanish Ministry of Science and Innovation: CGL2009-12926-C02-02, CGL2012-39850-CO2-02.

### Competing Interests

The authors declare there are no competing interests.

### Author Contributions

- Zaida Ortega performed the experiments, analyzed the data, wrote the paper, prepared figures and/or tables.
- Abraham Mencía performed the experiments, reviewed drafts of the paper.
- Valentín Pérez-Mellado conceived and designed the experiments, reviewed drafts of the paper.

### Animal Ethics

The following information was supplied relating to ethical approvals (i.e., approving body and any reference numbers):
    Lizards were sampled under licences of the Castilla y León Environmental Agency (EP/CYL/320/2012).

### Data Availability

    The raw data has been supplied as Supplemental Information.

### Supplemental Information

Supplemental information for this article can be found online at http://dx.doi.org/10.7717/peerj.2085#supplemental-information.

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
