# Peer review of "Are mountain habitats becoming more suitable for generalist than cold-adapted lizards thermoregulation?"

_PeerJ, doi:10.7717/peerj.2085_

## Round 0.1 · original submission · Major Revisions

We obtained only one review for this paper. However, the review is quite detailed and informative. I went through the paper myself, and I provide my own comments below. My guess is that revision of the paper with suitable detailed point-by-point answers could be quite challenging, but I am ready to reconsider the manuscript, if the authors believe they can provide the appropriate responses.

The introduction is relatively short and presents a good overview of the subject. However, some sentences need to be revised (e.g. lines 39–43, which are really not easy to understand in their present version). The authors should also specify that they studied “behavioural thermoregulation” when presenting their objectives (line 79).

Was there any effect of the experimental device on position of the animals in absence of thermal gradient (e.g. corner effect)? Devices used to measure preferences have all their own biases that should be taken into account while analyzing the results. Was this taken into consideration?

Much of the “Data analysis” subsection is centred on the use of three indices that are only presented in one small paragraph at the end of the Results section. Description of these indices should be in a former sub-section and the “Data analyses” section should simply refer to statistical analysis. If the statistical analyses are described in the appropriate way, there is no need to indicate which statistical test was used following each data set. There are some adjustments in the writing to be done.

The values as such do not have to be reported in the text when the results are presented in Figures. The results of a posteriori tests should be presented directly on graphs or in Tables using appropriate letters or asterisks. Suitable information should then be added in captions. The authors should remind that Figures or Tables should stand alone. Of course, details do not have to be repeated in the text, avoiding information redundancy.

When data do not meet criteria for parametric analysis, they cannot be presented using mean ± s.e. as these parameters are not representative of the data distribution. Median box-plot graphs should be used.

Preferences obtained in the lab are often not representative from distribution in the wild for many ectotherm species. The real meaning of preferences vs direct body temperature measurements in the wild should be discussed in a more appropriate way. For example, most of body temperature presented in Fig. 6 for I. galani are quite higher than the “preferred” range. Could it be related to a bias associated with the experimental device? Is using the preference as the “reference” is the best approach to be used? Obviously, this needs to be discussed.

Lines 86 to 90: This information should be provided in the Introduction.

Lines 95 to 99: All these numbers should be presented in a Table.

Lines 134–136: Was there any behavioural effect of measuring cloaca temperature every hour? How was this done without disturbing the animals?

Figure 2: Identify the different panels of Fig. 2. Ranges for I. galani are very difficult to see as most of the time only one black line is visible. The authors should be also be more specific lines 171–177 and refer to appropriate panels for each of the four groups. The results of a posteriori tests (homogenous groups) should be indicated either in the Figure caption or using appropriate letters associated with each panel.

Figure 3: The results of a posteriori tests should be indicated on Fig. 3. Table 1 is then unnecessary.

Lines 213–220: It is very difficult for the reader to visualize the results here. The use of ANCOVA should be indicated in the “Data analysis” subsection with the mention that homogeneity of slopes was checked first before proceeding to the ANCOVA. This will make the reading of the Results far less tedious.

Reviewer 1 ·

Basic reporting

The language quality is generally acceptable. However, grammar and style editing by native speaker will be necessary prior to publication.
The Introduction section should contain more general information. The mentioning of the study system in the first paragraph is not the best strategy for a multidisciplinary journal.
Some figures are superfluous and could be discarded. Some legends and axis titles did not conforn to terms used in the main text.
For details, see comments to authors.

Experimental design

The estimation of the preferred body temperature range miss some important details, e.g. whether only surface-active lizards were measured.
The measurement of microhabitat availability violates the randomness assumption.
See comments to authors for details.

Validity of the findings

I am afraid, estimates of preferred body temperatures seem flawed in one species, which complicates interpreation of thermoregulatory indices, and thus the whole study.
Thermal specialization and generalization cannot be inferred from Tp ranges. It requires physiological support, i.e. measuring thermal perormance curves.
In the Discussion, results are frequently misinterpret. See detailed comments.

Additional comments

In this study, authors compare behavioral thermoregulation between two syntopic species of lacertid lizards. Both species markedly differ in their distributional ranges, and so authors assume that they also differ in their thermal requirements and thermoregulatory abilities. Given the time and energy demands, field thermoregulatory studies are always a valuable contribution to the field of thermal ecology. I am afraid, this study suffers from several, mostly methodological deficiencies. 1. The range of preferred body temperatures (Tp) in I. galani. In comparison with the distribution of field body temperatures, the Tp range is too narrow. There are two explanations for this result. First, wild lizards, for some reasons, maintain their body temperatures above their target range. This is less likely because lizards usually avoid temperatures above their preferred range. In addition, many studies have demonstrated that their performance steeply drop at these “suboptimal” temperatures. Second, preferred body temperatures were influenced by some unknown factor. For example, active lizards may maintain higher and more variable temperatures than lizards in their refuge. Authors did not discriminate between active and non-active individuals. Because most variation in thermoregulatory indices resulted from the Tp differences between species, most of their findings is difficult to interpret.
2. The availability of microhabitats. Authors estimated the “random” availability of microhabitats within 1 m around lizard’s sight. This approach violates the randomness assumption, because lizard’s occurrence within a study are is already affected by its microhabitat preferences. In addition, air and substrate measurements are insufficient descriptives of lizard’s operative temperature, and so analyses of such data are meaningless.

I am sorry for being so negative. I suggest to you discarding problematic passages, re-measuring Tp range in I. galani (if possible), and publishing your valuable data in a shortened format. Below, I provide specific comments and suggestions, which you can find helpful for your task.

31-59 The first paragraph seems to long. Try to divide it into two. For example, separate general introduction from the mountain lizards issue…

56 “thermal biology” is too general term. “thermal ecology” is more appropriate here. Check throughout the text.

l. 60-70 This paragraph could be deleted without lost of relevant information.

95 Standard deviations are more informative descriptives than SE’s here.

102 “seasonal” instead of “weather”

108 “Randomly” means that you numbered all microhabitats and than randomly chose subsample to place your models here. Is this correct? If not replace “randomly” with “haphazardly”. Importantly, were the selected microhabitats within the home range of lizards in both species?

112-113 What are air and substrate temperatures good for? Please specify.

113-114 Did you calibrate digital thermometer and dataloggers?

115 What is the height of microhabitat? Please specify.

120 I am afraid these points are not random, because they are influenced by lizard microhabitat use.

134-135 Selected or preferred body temperature? Be consistent throughout the text. In addition, Tsel and Tpref are more suitable abbreviations for selected and preferred body temperatures, respectively, than Tset.

135-136 Because you measured body temperatures of surface-active lizards in the field, you should also measure active lizards in the gradient. Did you consider activity in your measurements?

137-140 Because you have no strong argument for using the central 50% as the Tp range, I suggest to calculate thermoregulatory indices using the 80% range. The reason for this is evident in Fig. 6. Clearly, the restricted range poorly characterizes selected body temperatures. See also comments below.

153-154 Unfortunately, the use of ratios in this formula may produce misleading results. Simple difference between de and db is a more reliable alternative (see Blouin-Demers and Weatherhead 2001).

156 Provide information about the accessibility of used software.

159 What kind of bootstrap the program calculates? Given the problematic distribution of used indices the bootstrap has to be non-parametric.

163 Given the issues above mean and 95% CI’s calculated using non-parametric bootstrap are more appropriate at least for thermoregulatory indices.

175 Because lizards frequently use hot microsites for basking, operative temperatures above the Tp range do not necessarily indicate thermally unsuitable microhabitats. Just the opposite, they enable to accelerate heating rates, and thereby reduce time spent basking. Please use another term.

177 Because the major aim of your paper is to examine if mountain habitats are increasingly unsuitable for mountain lizards, it would be informative to provide proportions of operative below, within and above their Tp ranges in both species. Moreover, because both species seem to use different substrates, it would be interesting to compare these proportions with respect to their predominantly used microhabitats. This provides some indication if substrate use buffers high environmental temperatures.

182 I guess that Ts is highly correlated with Ta. So, what temperature affects lizard’s microhabitat choice? Given the concept of operative temperature, these results does not make sense to me. Perhaps, more arguments supporting measurements of air and substrate temperatures would help in the MM section. Or consider deleting this part. See also comments above.

195 The measuring of microhabitat availability was not random. Check throughout ms.

202-208 You refer to the “preferred body temperature range” in the text, while you provide mean Tp in brackets.

203 Please round your temperature values to the resolution of your thermometer, i.e. 0.1°C. Check throughout the text.

211-212 Although species differences are obvious, heteroscedasticity of variances indicates that the assumption of Mann-Whitney U test was violated. Please note that both parametric and non-parametric tests require homogeneous variances between samples.

214 Why did you use ANOVA to compare two samples? Did you examine the Tb variation among seasons before clumping all temperatures together?

214-220 The association between Tb and environmental temperatures reminds old approaches to study thermoregulation. Because this approach may provide misleading results about behavioral thermoregulation, I recommend deleting the problematic part. Otherwise, provide more rationale for this.

221 Delete “value of the”.

227-228 Unclear, please reformulate.

229-230 In comparison with field body temperatures the Tp looks strange. It appears that lizards maintained the most body temperatures above its target range. Why? The most parsimonius explanation is that the estimation of Tp range was affected some unknown factor. Be careful with its interpretation.



233-238 Thermal specialists have narrower thermal performance curves (TPC) than thermal generalists. Because you did not measure TPCs and you have no information about the relationship between TPCs and the Tp range, this statement is preliminary. The formulation should be softened.

242-250 Given the arguments above, this information has little relevance to your study. Discuss your results, please.

250-253 I am afraid, you have no data to support this conclusion. The extremely narrow Tp range looks suspicious. You should discuss the difference between the Tp range and the distribution of field body temperatures.

256-258 Yes, may be. But your Te measurements suggest that that the risk of overheating is not a current issue, because lizards has still some proportion of microsites within and below their Tp range. So, it appears that behavioral adjustments can easily buffer the impact of changing climate in your system.

269-271 It would be nice to mention the possible role of acclimation of Tp range in coping with climate change.

276 air temperature

277 Your results suggest the opposite. Given the species specific substrate use, I. galani has higher availability of suitable substrates than P. bocagei. From a thermal view, rocky substrate provides lizards with the range of temperatures, which allow effective thermoregulation.

277-280 Fig. 6 clearly show that the variation in E resulted from disparate Tp ranges rather than from the availability of operative temperatures. As I mentioned above the Tp range of I. galani does not look as the target range for lizard thermoregulation in the field.

Fig. 1. Delete “Worldwide”.
Fig. 2. Provide information about used descriptives. The Tp range are not easily discernible.
Fig. 3 Informative value of this graph is problematic (see comments above). Consider deleting it.
Fig. 4. Microhabitat use in two lizard species, I. galani and P. bocagei. Note that it is preferred to provide not relative but absolute values of observations.
Fig. 5. Information provided in this graph can be easily placed to the text. Distributions of preferred body temperatures (more informative than boxplots) are presented in Fig. 6. So consider deleting this figure. If not, be consistent with the used terms, i.e. preferred vs. selected body temperatures.
Fig. 6. Temperature distributions in two lacertid lizards. Again, be consistent with the terminology used. As I mentioned above, the Tp range of I. galani is problematic

Table 1. Similarly as in Fig. 3. Consider deleting.

Table 2. SE’s are too low, because you calculated this descriptive from the distribution of randomized values, right? I strongly suggest to calculate mean db and de from raw data and provide all indices with 95CIs obtained using non-parametric bootstrapping technique (i.e. “boot” package in R).

---

## Round 0.2 · Minor Revisions

The authors submitted a revised version with sound answers to the different comments sent with the first evaluation. I would suggest very minor corrections to be made before final acceptance of the manuscript.

1. Line 41: “seem” instead of “seems” (plasticity and evolution … seem)
2. I would not modify text line 64, but I would move the reference to Table 1 in the M&M section. In other words, in the introduction you indicate that the two species are medium-size species, but in the M&M section you should indicate that length and weight of animals used in the study are provided in Table 1.
3. Lines 258–260: I understand that this sentence was added to answer one of the comments of the reviewer. However I do not understand how there can be an “acclimation of the thermal preferences”. Animals may change their preference while acclimating to new temperature conditions, but they do not acclimate their preference (?). I would suggest rephrasing the sentence.
4. Caption of Fig 2 “Non-significant post-hoc results are marked in the panels with the same letter”. I am not sure to understand what you mean. Would it be correct to state that “The different letters indicate significant different operative temperatures among microhabitats”?
5. Table 2 “All microhabitats were measured under all sun situations, so it is homogeneous among them, making the proportions comparable.” I had difficulties with this sentence. What “it” refers to? I would suggest to rephrase this.

---

## Round 0.3 · accepted · Accept

Thank you for this revised version.